# *Pseudomonas aeruginosa* Resistance to Bacteriophages and Its Prevention by Strategic Therapeutic Cocktail Formulation

**DOI:** 10.3390/antibiotics10020145

**Published:** 2021-02-02

**Authors:** Andrew Vaitekenas, Anna S. Tai, Joshua P. Ramsay, Stephen M. Stick, Anthony Kicic

**Affiliations:** 1Occupation and the Environment, School of Public Health, Curtin University, Perth, WA 6102, Australia; 2Wal-Yan Respiratory Research Centre, Telethon Kids Institute, The University of Western Australia, Crawley, WA 6009, Australia; andrew.vaitekenas@telethonkids.org.au (A.V.); Stephen.Stick@health.wa.gov.au (S.M.S.); 3Department of Respiratory Medicine, Sir Charles Gairdner Hospital, Perth, WA 6009, Australia; Sze.Tai@health.wa.gov.au; 4Institute for Respiratory Health, Perth, WA 6009, Australia; 5Faculty of Health and Medical Sciences, The University of Western Australia, Perth, WA 6009, Australia; 6Curtin Medical School and Curtin Health Innovation Research Institute, Curtin University, Perth, WA 6102, Australia; josh.ramsay@curtin.edu.au; 7Division of Paediatrics, School of Medicine, The University of Western Australia, Perth, WA 6009, Australia; 8Department of Respiratory and Sleep Medicine, Perth Children’s Hospital, Perth, WA 6009, Australia; 9Center for Cell Therapy and Regenerative Medicine, School of Medicine and Pharmacology, The University of Western Australia and Harry Perkins Institute of Medical Research, Perth, WA 6009, Australia

**Keywords:** phage resistance, bacteriophages, *Pseudomonas aeruginosa*, cystic fibrosis, phage therapy

## Abstract

Antimicrobial resistance poses a significant threat to modern healthcare as it limits treatment options for bacterial infections, particularly impacting those with chronic conditions such as cystic fibrosis (CF). Viscous mucus accumulation in the lungs of individuals genetically predisposed to CF leads to recurrent bacterial infections, necessitating prolonged antimicrobial chemotherapy. *Pseudomonas aeruginosa* infections are the predominant driver of CF lung disease, and airway isolates are frequently resistant to multiple antimicrobials. Bacteriophages, or phages, are viruses that specifically infect bacteria and are a promising alternative to antimicrobials for CF *P. aeruginosa* infections. However, the narrow host range of *P. aeruginosa*-targeting phages and the rapid evolution of phage resistance could limit the clinical efficacy of phage therapy. A promising approach to overcome these issues is the strategic development of mixtures of phages (cocktails). The aim is to combine phages with broad host ranges and target multiple distinct bacterial receptors to prevent the evolution of phage resistance. However, further research is required to identify and characterize phage resistance mechanisms in CF-derived *P. aeruginosa,* which differ from their non-CF counterparts. In this review, we consider the mechanisms of *P. aeruginosa* phage resistance and how these could be overcome by an effective future phage therapy formulation.

## 1. Introduction

The increased rate of antimicrobial resistance in human and animal pathogens presents the prospect of a post-antibiotic era, prompting the World Health Organization (WHO) to compile a list of priority antimicrobial-resistant pathogens requiring the development of alternative antimicrobials [1]. Despite the urgency, economic hurdles have led to a paucity in the development of new antimicrobials, leaving fewer drugs to treat priority multi-resistant pathogens such as *Acinetobacter baumannii*, *Pseudomonas aeruginosa*, and Enterobacteriaceae [2,3]. Without drastic action to combat antimicrobial-resistant infections, they are predicted to be the primary cause for up to 10 million deaths globally by 2050 and the associated loss in gross domestic product is USD 100 trillion [4]. The burden of antimicrobial-resistant infections will be greatest in individuals with chronic conditions, as they are disproportionately affected by bacterial infections [5]. Amongst these are people with cystic fibrosis (CF), who are predisposed to recurrent and persistent bacterial infections. CF is caused by mutations in the Cystic Fibrosis Transmembrane Conductance Regulator (*CFTR*) gene. Mutations result in dehydration of the air surface liquid on airway epithelial cells [6], leading to defective mucociliary clearance and the concentration of viscous mucus within the airways [7]. The viscous and stagnant mucus of the CF lung microenvironment is conducive to bacterial attachment and propagation [8] and ensuing bacterial infections stimulate inflammation, irreversible structural lung changes, damage, and functional decline in the lungs, collectively known as progressive lung disease [9,10,11]. Prolonged and recurrent bacterial infections lead to respiratory failure, which is the most common cause of death in those with CF [12,13].

## 2. *Pseudomonas aeruginosa* and CF

*P. aeruginosa* is widely considered the most important bacterial infection in CF, as its associated inflammation drives lung disease progression and, ultimately, mortality [10,11,14,15,16,17,18]. Infections begin with a period of intermittent isolation in children, typically 2–3 years of age, but can occur much earlier in life, even by 3 months of age [19,20]. Early intermittent colonisations are typically caused by environmental *P. aeruginosa* isolates [21], which commonly exhibit a non-mucoid colony morphology and are usually motile and virulent but still susceptible to antimicrobials [22,23]. However, widespread *P. aeruginosa* strains (epidemic strains) can also be significant sources of infection and antimicrobial resistance [24,25,26,27,28,29,30]. Eradication of *P. aeruginosa* following each infection is attempted to prevent chronic colonisation and airway damage [31,32,33,34,35,36,37]. Nevertheless, vigilant detection, aggressive eradication, and infection control strategies are successful in delaying, but not preventing, the eventual establishment of chronic *P. aeruginosa* in the CF airway [13,38].

Once a CF patient is persistently colonised with established *P. aeruginosa* populations, they often form antimicrobial-resistant biofilms [39]. Additionally, despite arising from a clonal population, persistent *P. aeruginosa* populations diversify through mutation [40]. The diversification process is frequently accelerated via hypermutable sub-populations of cells with deficits in DNA repair systems [41,42,43,44]. However, features typically exhibited by persistently infecting strains include high antimicrobial resistance, mucoid colony morphology, and attenuated motility and virulence [45]. These adaptations likely contribute to *P. aeruginosa* being the most common persistently colonising pathogen in adults with CF, where 80% are chronically infected [38]. To treat antimicrobial-resistant infections, clinicians must use last resort antimicrobials, cocktails, and high drug concentrations that present safety issues [46,47]. Current strategies are not viable long-term solutions for CF treatment in the post-antibiotic era and alternative antimicrobials are urgently required.

## 3. Phage Therapy

Bacteriophages (phages) are viruses that specifically infect bacteria and present an alternative treatment strategy to antimicrobials. Phages bind to specific receptors on the bacterium surface, inject their genome, self-replicate using the host cell machinery, and lyse the bacterium to release progeny [48]. Phages have a long history of use, especially in Eastern European countries where early proof of principle trials were performed and phage therapy is still practised today [49]. Despite the early trials of phage therapy often being successful, there were also notable treatment failures due to technological and knowledge limitations [50], leading to poorly manufactured phage therapies [51].

There are many approaches to phage therapy including single phage, phage cocktail, phage enzyme, and combination therapies, which have been reviewed previously [52]. Before a therapy is produced, phages must be isolated and characterised. Phages are isolated where their host bacteria are found [53,54]. Common sources include clinical infection samples, like sputum [55], or from wastewater, especially if it is collected from around hospitals [56]. Isolation of phages active against different bacterial species have varied in success, but *P. aeruginosa* phages tend to be the most frequently isolated [57]. Characterisation of isolated phages is essential to ensure that they exclusively exhibit a bactericidal lifecycle, are active against clinically relevant bacteria, and do not contain bacterial genes that could improve a bacterium’s fitness [53,54].

Advances in synthetic biology have allowed the design of phages from previously isolated and rigorously characterised phages. These engineered entities can diversify phages available for a host bacterium without further isolation [58]; augment a phage’s host range [58], including across species and genus levels [59]; prevent phage resistance [60]; and alter the action of phages that could increase their safety to humans [61]. Phage engineering has primarily used homologous recombination of a phage’s genome with scaffolds in a bacterial host or the assembly of a complete phage genome from small fragments [62,63]. Genetic engineering through recombination has been hampered by the low recombination rates of phages [62,63]. However, the natural recombination sites of temperate phages have been leveraged to increase the efficiency of the process [62,63]. To identify recombinant mutant phages, extensive screening is thus required, which can be aided with a selection stage [62,63]. Yet, assembling small fragments into a phage genome does not necessarily require a bacterial host and, therefore, can use genes that are toxic to a bacterium and can be performed at a higher efficiency for lytic phages [62,63]. It is currently difficult to assemble large genomes and protocols are complicated or still being established for many host bacteria [62]. Genetically engineered phages offer commercialization opportunities that natural phages do not [62], but they may face extra regulatory challenges [51].

Regardless of their source, once they pass the characterisation stages of research, phages must be produced according to Good Manufacturing Practice (GMP) [51]. However, large-scale production processes can be complicated due to the biological nature of both phages and their bacterial host [64] and the lack of a complete set of phage-specific GMP guidelines [51]. There is a particular paucity in GMP guidelines for the quality control of phage therapeutics, but this can be informed by readily available information including host range, host bacterial component contamination, morphology through transmission electron microscopy, and genome sequence analysis [51].

## 4. Phage Therapy for the Treatment of *P. aeruginosa*


Numerous studies have demonstrated phage therapies’ effectiveness against *P. aeruginosa* and eukaryotic tolerability in in vitro and in vivo models [65,66,67,68,69,70,71,72,73,74,75,76,77,78]. Individual phages only infect a specific subset of *P. aeruginosa* strains, so the effectiveness of phage therapy may vary greatly depending on the specific CF lung population. Therapy effectiveness can be improved by combining phages with a diversity of host ranges into cocktails [65,76] and synergistic phage-antimicrobial cocktail therapy [71,75,79,80]. Initial experimental successes have led to the compassionate use of phage therapy for treatment of antimicrobial-resistant infections, including *P. aeruginosa* in those with CF, where phages have been identified as effective and safe [49,81,82,83]. However, to be more widely translated, clinical trials are required. There have been two clinical trials of phage therapy: a trial involving the treatment of *P. aeruginosa* ear [84] and burn infections [85]. Phage therapy was shown to be effective and safe in the relatively small trial against *P. aeruginosa* ear infections [84]. Whereas in the trial for the treatment of burn wound infections, only safety was demonstrated, since there were issues manufacturing GMP-quality phage cocktails at sufficient titre [85]. Inconsistent results necessitate additional clinical trials, not only specifically in this context, but also for the treatment of other infections, such as *P. aeruginosa* CF lung infections. To enable clinical trials and for phage therapy to become a standard adjunct to antimicrobials in CF, further detailed pre-clinical studies are needed.

One area requiring further investigation is the evolution of phage resistance and its suppression [86]. The importance of phage resistance has also been realised in the compassionate treatment of antimicrobial-resistant infections and necessitated rapid cocktail reformulation to prevent treatment failure [87,88,89,90]. As a result, it is important to consider *P. aeruginosa* phage resistance to ensure the efficacy of phage therapy in the long-term.

## 5. Phage Resistance

Bacterial resistance to phages can occur at every stage of the phage lifecycle from: (i) receptor recognition and binding, (ii) genome injection, (iii) DNA replication, (iv) transcription and translation, (v) phage assembly, and (vi) phage release. The general bacterial phage-resistance mechanisms have been reviewed previously [91] and are still being discovered [92]. The evolution of bacterial resistance to phages is very rapid due to there being billions of bacteria in a single colony and the existence of resistant mutants. Likewise, phage evolution in response to bacterial changes is also rapid because 10 to >100 phages are released by productive infection of a host bacterium. The rapid reciprocal evolution of bacteria and phages has been explained by the Red Queen theory of evolution, where parasite and host are continually mutating, resulting in the relationship appearing static [93]. In the pursuit of understanding this host–parasite relationship, two of the most important genetic engineering technologies have been discovered, namely Clustered Regularly Interspaced Short Palindromic Repeat (CRISPR-cas) and restriction enzymes. However, in the context of phage therapy, the resistance predominantly occurs at the receptor recognition and binding stage through receptor mutations [94].

## 6. *P. aeruginosa* Phage Resistance

### 6.1. Receptor Mutations

Commonly identified receptors for *P. aeruginosa* phages are lipopolysaccharide (LPS) and type IV pili (T4P) [95], with the phage OMKO1 using OprM, which is a key component of the MexAB-OprM antibiotic efflux pump [96]. Development of resistance to OMKO1 was shown to increase antimicrobial susceptibility in an initially multi-drug-resistant strain of *P. aeruginosa* [96]. This suggests that the effect of phage resistance may be beneficial in reducing antimicrobial resistance and that combined antimicrobial-phage therapy could prevent resistance evolving to either treatment. The phage resistance mutations identified in previous studies have been summarized in Appendix A.

LPS is composed of lipid A (membrane embedded domain), a core (domain connecting lipid A and O-antigen), and O-antigen (polysaccharide that extends extracellularly). *P. aeruginosa* expresses three types of O-antigen, which include the common polysaccharide antigen (CPA; homopolymer O-antigen), O-antigen specific (OSA; heteropolymer O-antigen), and uncapped antigen (no or one O-antigen sugar). Therefore, resistance mutations for LPS-exploiting phages vary according to which LPS component or type the phage specifically binds (Figure 1). Mutations to T4P can occur in genes encoding structural components, therefore preventing the receptor being created or those that drive pilus retraction for twitching motility (Figure 1). Retraction of T4P is required by pili-specific *P. aeruginosa* phages because it allows them to meet the host membrane and continue infection [97]. Additionally, receptor mutations range from single base-pair alterations to >100 kbp multi-gene deletions (Appendix A). Receptor mutations can also have wider effects than the phage receptor through altering genes with multiple functions (Appendix A). In some cases, large or regulatory gene mutations may be beneficial to *P. aeruginosa* by providing cross-resistance to other phages, they can also result in increased fitness costs [98]. Oechslin et al. (2016) found that a phage-resistant mutant carrying a large genomic deletion had increased susceptibility to ciprofloxacin and by exploiting this via combined phage-antimicrobial therapy, resistant bacterial mutants were suppressed. However, even small mutations to phage receptors, when phages are not present, have been shown to attenuate bacterial growth [78,99], biofilm formation [100,101], motility [71,78,101,102], antimicrobial resistance [71,96], virulence, and infectivity [71,78,103].

The strength of both phage resistance and the associated fitness cost is phage-specific [99]. Therefore, it is theoretically possible to combine phages into cocktails to provide multiple distinct selection pressures, which also comes with a maximum cost to fitness. Yang et. al. (2020) strategically formulated a cocktail of phages that successfully suppressed the evolution of resistance, but after prolonged incubation phage resistance was observed. Further suppression of phage resistance could be attained by using phages specific for more receptors, combining phages with antimicrobials, exploiting the fitness cost of phage resistance in therapeutic formulation, and including phages that kill specific resistant mutants of bacteria (escape mutants; Figure 2a). Moreover, the use of strains from non-CF origins in all but one of the studies of *P. aeruginosa* phage resistance [71] has neglected the contributions of the unique adaptions of CF clinical isolates of *P. aeruginosa* on phage therapy and resistance. Therefore, this paucity limits the relevance of findings to the treatment of CF airway infections.

### 6.2. Superinfection Systems

Temperate phages, capable of both lysis and integration into the host bacterial genome, are abundant in CF isolates of *P. aeruginosa* [104,105]. Phages exhibiting temperate lifecycles are not used for phage therapy because integration into the host genome can increase *P. aeruginosa* virulence [104,106], antimicrobial resistance [104], and biofilm formation [105]. Temperate phages integrated into bacterial genomes can also increase host fitness by providing resistance to the subsequent phage infection, known as superinfection resistance. Mechanistically, this is achieved through the alteration of the two most common *P. aeruginosa* phage receptors (T4P and LPS) and production of repressors of phage infection [107]. Superinfection resistance prevents the infection of a limited range of phages, which is determined by both the temperate phage and the *P. aeruginosa* host [107,108,109,110]. As phage therapy does not use temperate phages, superinfection resistance can be circumvented by conducting host range experiments. The host range experiments enable the identification of lytic phages that are not resisted by the temperate phages contained in a *P. aeruginosa* isolate. However, due to the mobile nature of temperate phages, these may become a problem in vivo, as they may transfer superinfection resistance between different *P. aeruginosa* strains. However, the chance of temperate phage mobility causing treatment failure can be minimised with an adequately diverse cocktail as superinfection resistance is only provided to a defined range of phages [107].

### 6.3. Masking Phage Receptors

The phenomenon of masking has been shown to be mediated by glycosylation of the T4P receptor on *P. aeruginosa,* providing it with resistance [111]. Unlike mutations to phage receptors, glycosylation does not affect the T4P-mediated twitching motility [111]. To overcome this mechanism of resistance, phages identified during host range characterisation could be selected or trained to be capable of infecting bacteria with a glycosylated T4P. Moreover, *P. aeruginosa* formation of biofilms and secretion of exopolysaccharide (EPS) can facilitate masking of receptors by preventing phage penetration and receptor binding [112]. However, some phages are more capable of penetrating biofilms due to the production of matrix degradative enzymes [113].

### 6.4. CRISPR-Cas

CRISPR-cas arrays are abundant in *P. aeruginosa* including CF isolates [114] and can mediate immunity to specific phages [115]. The CRISPR RNA (crRNA) component of the CRISPR-cas binds complementary phage-specific DNA sequences allowing cas nucleases to cleave the recognised invader’s DNA. CRISPR-cas is also capable of adaptation so that a bacterium that survives an infection can incorporate a phage’s DNA into its CRISPR-cas array. Adaptation allows a bacterium to recognise and prevent subsequent infections with the same phage. The CRISPR-cas found in *P. aeruginosa* falls into the subtype I-F, which has four cas-associated genes (that aid crRNA production and target recognition) and two cas nuclease genes (for cleavage of target and adaptation) [114]. However, phages are able to resist CRISPR-cas resistance [116,117] and therapy uses cocktails of diverse phages, meaning it is unlikely to be the main form of resistance that evolves [94]. Nonetheless, recent in vitro work has shown that *P. aeruginosa* isolates develop CRISPR-cas resistance to single phages [94].

### 6.5. Restriction-Modification (RM) Systems

Restriction-modification (RM) systems also provide immunity to phages by recognising and cleaving DNA sequence motifs [118]. Unlike CRISPR-cas systems, these motifs are less phage-specific and not adaptive. To prevent destruction of the bacterial genome, the nucleases are often associated with methylases or other nucleotide modifiers that mark the motifs on the self-DNA, thereby preventing its cleavage [118]. Two RM-associated systems have recently been characterised including Bacteriophage Exclusion (BREX) and Defence Island Associated with Restriction-Modification (DISARM) [92]. A third system called DNA Degradation (DND) denotes self-DNA via sulphur addition [119]. Identification and characterisation of these RM-associated systems and their role in *P. aeruginosa* have not been thoroughly explored. Although genome analysis has indicated BREX and DISARM components are present in *P. aeruginosa* genomes [92]. Similarly, other phage resistance mechanisms, like cell-signalling coupled with abortive infection and many unknown modes of action systems, have been predicted by *P. aeruginosa* genomic analysis, but are yet to be functionally verified [92].

## 7. Phage Evolution

As previously mentioned, the relationship between bacteria and phages is reciprocal and the ability of phages to evolve alongside bacteria is an advantage compared to conventional antimicrobials. During DNA replication, phages undergo small mutations (Figure 2b). These are often in the receptor binding proteins, which are responsible for phage-host range and receptor recognition [120]. When multiple phages are present, as in a cocktail, evolution can occur through genome recombination (Figure 2c). Recombination creates a phage with a mosaic genome of its multiple ancestors (Figure 2c) [121]. Experimentally, evolution through recombination can produce a phage capable of infecting several initially resistant *P. aeruginosa* strains [121]. Evolution of phages can also be used to pre-adapt them to the bacterial host, suppressing the propagation of both receptor mutations [122,123] and CRISPR-cas resistance [115]. Additionally, phages can be evolved (known as training) to overcome the resistance evolved by a *P. aeruginosa* strain [124]. Experimentally evolved phages require stringent genome characterisation to identify any that have picked up bacterial or temperate phage genes. Phages found to contain any deleterious genes can then be excluded from being considered in therapeutic formulation.

Phages have also evolved genes that encode proteins capable of overcoming other resistance mechanisms employed by *P. aeruginosa*. To overcome the resistance mediated by the masking of receptors, phages have evolved to carry biofilm-degrading enzymes [113]. The enzymes breakdown the exopolysaccharide molecules in a biofilm matrix and allow more efficient penetration and phage infection [113]. Therefore, isolating and identifying phages with biofilm-degrading enzymes, through genome bioinformatic characterization, could provide preferential constituents of a cocktail for the treatment of *P. aeruginosa* biofilm infections.

Additionally, phages can carry anti-CRISPR genes whose products circumvent CRISPR-cas resistance. Anti-CRISPRs can function through binding CRISPR-cas components, causing conformation changes that are non-functional; mimicking DNA to competitively bind CRISPR-cas, preventing recognition; binding cas nucleases, making them unable to interact and degrade phage sequences; and silencing the expression of CRISPR-cas genes [125]. Phages found to contain anti-CRISPRs, during characterisation, may be more applicable for in a therapeutic formulation to prevent CRISPR-cas resistance.

To prevent RM system resistance, phages modify their nucleotide bases. Modified nucleotides in phage genomes, such as archeosine [54], queosine [126], hypermodified thymidine, and guanine bases [127], prevent DNA motif recognition and degradation. Phages more resistant to RM resistance can be found through genomic analysis identifying those with genes capable of modifying nucleotides [54].

## 8. Conclusions

In summary, phage therapy is a promising alternative to antimicrobials in an increasingly post-antibiotic era. Specifically, this offers a solution for individuals with CF where antimicrobial-resistant infections with *P. aeruginosa* drive progressive lung disease. Although resistance to phages arises, phages can also counter-evolve to overcome the mechanisms employed by bacteria. However, more work is required to understand the evolution of phage resistance in CF clinical isolates of *P. aeruginosa* and what strategies can be implemented to overcome these potential barriers. Through understanding the spectrum of mechanisms affording phage resistance and their fitness trade-offs in a CF disease relevant model, phage therapies, capable of increasing effectiveness and suppressing resistance, can be strategically formulated. Such a step is crucial for phage therapy to translate to a standard adjunctive therapy, alongside antimicrobials in CF, without resistance emerging.

## Figures and Tables

**Figure 1 antibiotics-10-00145-f001:**
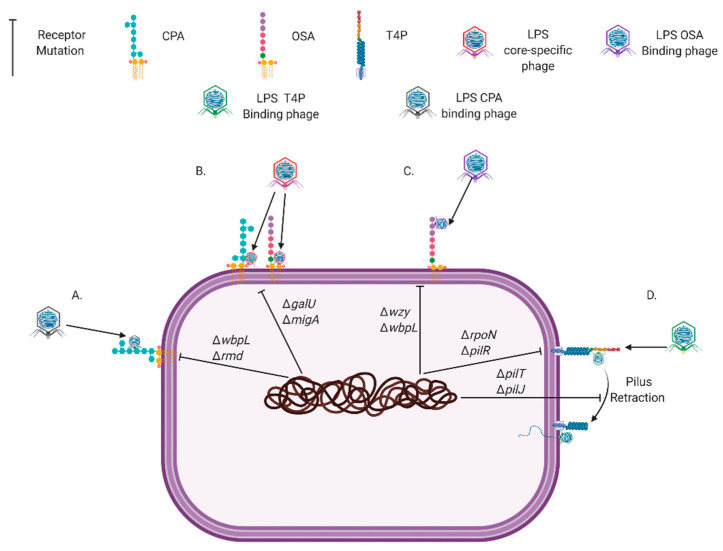
Common *Pseudomonas aeruginosa* genetic mutations that prevent the phage receptors Common Polysaccharide Antigen (CPA) (**A**), lipopolysaccharide (LPS) core (**B**), O-Specific Antigen (OSA) (**C**), and Type IV Pili (T4P) (**D**) being synthesized or functioning. Created with BioRender.com.

**Figure 2 antibiotics-10-00145-f002:**
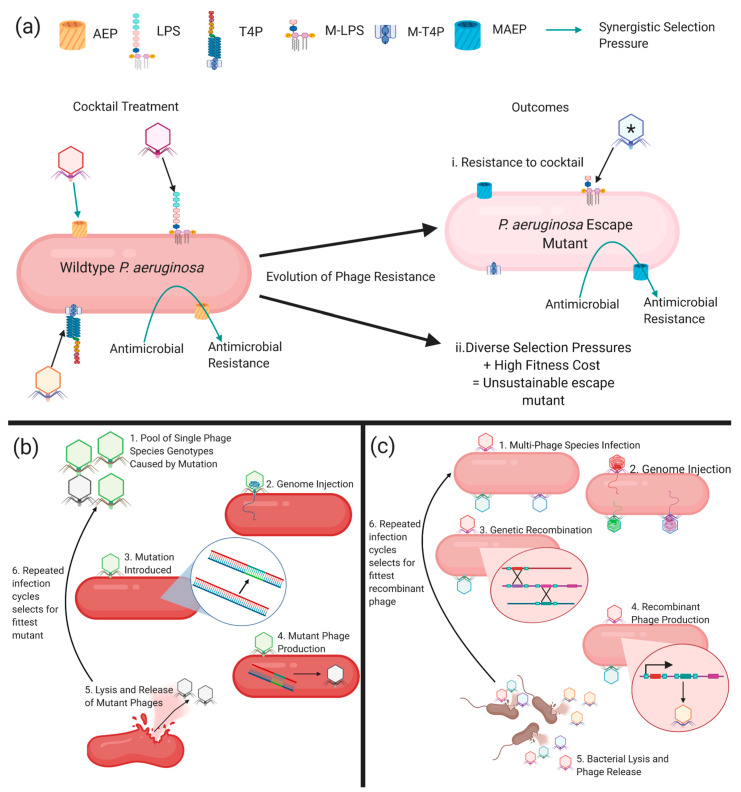
(**a**) The rationale for strategic cocktail formulation. Cocktails contain phages that use different receptors like lipopolysaccharide (LPS), type IV pili (T4P,) or antimicrobial-resistant determinants (antibiotic efflux pumps; AEP) so that their addition with antimicrobials provides opposite and synergistic selection pressures. This will result in outcome i, where resistance arises to the phage cocktail through mutations that alter the availability of phage receptors. For example, this outcome may result in a mutated LPS (M-LPS), a mutated T4P (M-T4P), and a mutated antibiotic efflux pump (MAEP) with conserved efflux function. This outcome can be prevented by adding the phage (denoted by *) to the cocktail that specifically kills the escape mutant, therefore forcing further different mutations in the bacteria, which could result in outcome ii, whereby the selection pressures are so diverse and come with a high fitness trade-off that resistance is unsustainable. (**b**) Phage evolution by spontaneous small mutation during DNA replication. (**c**) Phage evolution by genetic recombination between multiple phages. Created with BioRender.com.

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
