# Peer review of "Pseudomonas aeruginosa Resistance to Bacteriophages and Its Prevention by Strategic Therapeutic Cocktail Formulation"

_antibiotics, 2021, doi:10.3390/antibiotics10020145_

Round 1
Reviewer 1 Report
In the submitted manuscript, Vaitekenas et.al. summarized current knowledge of bacteriophages therapy to treat Pseudomonas aeruginosa infection. The authors first introduced P. aeruginosa, a major cystic fibrosis associated pathogen and concept of phage therapy. In addition, authors discussed mechanisms of phage-resistance in P. aeruginosa and proposed that phage cocktail therapy might overcome these phage-resistance mechanisms. Since phage cocktail therapy could potentially be utilized to treat P. aeruginosa infection in cystic fibrosis (CF) patients, this review will provide helpful information to the readers in the field who are interested in this topic. Overall, this review is comprehensive and well written.
Author Response
In the submitted manuscript, Vaitekenas et.al. summarized current knowledge of bacteriophages therapy to treat Pseudomonas aeruginosa infection. The authors first introduced P. aeruginosa, a major cystic fibrosis associated pathogen and concept of phage therapy. In addition, authors discussed mechanisms of phage-resistance in P. aeruginosa and proposed that phage cocktail therapy might overcome these phage-resistance mechanisms. Since phage cocktail therapy could potentially be utilized to treat P. aeruginosa infection in cystic fibrosis (CF) patients, this review will provide helpful information to the readers in the field who are interested in this topic. Overall, this review is comprehensive and well written.
Response 1: We thank the reviewer for their time in reviewing our submission, their positive and constructive feedback and their recognition in its value as a publication
Reviewer 2 Report
The review by Vaitekanas et al. gives a brief, though dense overview over the current successes and also problems and caveats in phage therapeutic approaches focussing on cystic fibrosis-associated Pseudomonas aeruginosa infections.
Due to the shortness and density of the text it is sometimes a little hard to follow the authors‘ intentions but together with the two figures it gives a good insight into the newest phage therapeutic developments also in combination with antibiotics.
There are only some minor concerns with the figure legends; also in paragraph 6 („phage evolution“) the importance of the co-evolution of bacteria and phage in terms of resistancies should be more clearly emphasized. The arms that phage raise against bacterial phage restistance strategies should be more explicitly mentioned. The important mechanisms of the phage’s anti-CRISPR immune answer, the antirestriction systems as well as cell surface/biofilm degrading phage enzymes are only mentioned in the respective bacterial sections (5.3-5.5). I suggest to put these important phage defense mechanisms that have been evolved explicitely into paragraph 6.
Figures:
Fig. 1
This figure is very nicely depicted and quite clear. However the 4 different targets of phage reception (CPA, LPS OSA and T4P) which are stated A, B, C and D in the figure legend should be found in the graphics itself to make these four distinct cases clearer. I suggest to insert the letters A-D in the lower part of the figure.
Fig. 2
This figure is rather complex and not every detail was clear to me. The two major points are: Is the blue phage in (a) the escape killer phage? And why is the mutant MAEP still functioning; what kind of mutant is meant here?
Table:
In the last column it would be more helpful to the reader to give the numbers of the references from the Refs section rather than the first author’s name.
Minor points:
Introduction:
line 41: …...human AND animal pathogens…….
line 54: ……air surface liquid VISCOSITY on airway……
paragraph 4. Phage-resistance
line 117: …..because 10 to > 100 phages PER HOST BACTERIUM are produced…..
paragraph 5. P. aeruginosa Phage-resistance
line 156: ….antibiotic efflux PUMP….
line 169/170: …because PILI-SPECIFIC phages require…..
paragraph 6. Phage Evolution
as stated in the beginning of this letter, the section 6 should put more emphasis on phage arms against bacterial phage resistance mechanisms
line 327/328: ….and supresses the PROPAGATION (instead of evolution) of receptor mutations [105, 106] and of CRISPR-cas resistance, RESPECTIVELY…..
paragraph 7 Conclusions
line 335 …a solution for those PATIENTS with CF where….
Line 337: p…phage arise, they can also COUNTER-evolve to overcome….
References
line 485-488: the two references # 51 and # 52 are identical, PLEASE CHECK
Author Response
Point 1: Fig. 1 is very nicely depicted and quite clear. However, the 4 different targets of phage reception (CPA, LPS OSA and T4P) which are stated A, B, C and D in the figure legend should be found in the graphics itself to make these four distinct cases clearer. I suggest to insert the letters A-D in the lower part of the figure.
Response 1: We thank the reviewer for their recommendation and have now edited Figure 1 accordingly (page 4).
Point 2: Fig. 2 This figure is rather complex and not every detail was clear to me. The two major points are: Is the blue phage in (a) the escape killer phage? And why is the mutant MAEP still functioning; what kind of mutant is meant here?
Response 2: We apologize if Figure 2 was not absolutely clear. The ‘blue phage’ queried by the reviewer is indeed the bacterial escape mutant. For additional clarity we have denoted it with a * and the legend modified (line 230) “This outcome can be prevented by adding the phage (denoted with a *) to the cocktail that specifically kills the escape mutant therefore forcing further different mutations in the bacteria which could result in outcome ii.”
Furthermore, in response to the second query, we again have edited the legend for clarity. Lines 227-230, “This will result in outcome i. resistance arising to the phage cocktail through mutations that alter the availability of phage receptors. For example, resulting in mutated LPS (M-LPS), mutated T4P (M-T4P) and mutated antibiotic efflux pump (MAEP) with conserved efflux function.”
We feel that these improvements now provide full clarity behind the figure’s purpose.
Point 3: In the last column it would be more helpful to the reader to give the numbers of the references from the Refs section rather than the first author’s name.
Response 3: We thank the reviewer for the suggestion and the citations in the reference column of Supplementary Table 1 have been modified to the number style as used in the body of the manuscript.
Point 4: line 41 …...human AND animal pathogens…….
Response 4: In response we have edited the text accordingly (line 38).
Point 5: line 54 ……air surface liquid VISCOSITY on airway……
Response 5: In response, we feel the text is appropriate. To clarify, the airway surface liquid is dehydrated as a result of the gene mutations that cause CF which in turn is known to increase mucus viscosity.
Point 6: line 117…..because 10 to > 100 phages PER HOST BACTERIUM are produced…..
Response 6: In response, the sentence has been amended to has been amended as proposed with further clarification “because 10 to >100 phages are released by productive infection of a host bacterium” (lines 158-150).
Point 7: line 156….antibiotic efflux PUMP….
Response 7: We now have included the excluded word as indicated (line 176).
Point 8: line 169/170 …because PILI-SPECIFIC phages require…..
Response 8: The sentence has been modified as proposed (line 190-191).
Point 9: As stated in the beginning of this letter, the section 6 should put more emphasis on phage arms against bacterial phage resistance mechanisms.
Response 9: We thank the reviewer for their constructive feedback. In response, information about phages’ response to bacterial resistance, that was contained in various other sections has now been move to section ; lines 319-330, “Phages have also evolved genes that encode proteins capable of overcoming other resistance mechanisms employed by P. aeruginosa. To overcome the resistance mediated by masking of receptors, phages have evolved to carry biofilm degrading enzymes [113]. The enzymes breakdown the exopolysaccharide molecules in a biofilm matrix and allow more efficient penetration and phage infection [113]. Therefore, isolating and identifying phages with biofilm degrading enzymes, through genome bioinformatic characterization, could provide preferential constituents of a cocktail for the treatment of P. aeruginosa biofilm infections.
Additionally, phages can carry anti-CRISPR genes whose products circumvent CRISPR-cas resistance. Anti-CRISPRs can function through: binding CRISPR-cas components, causing conformation changes that are non-functional; mimicking DNA to competitively bind CRISPR-cas preventing recognition; bind cas nucleases making them unable to interact and degrade phage sequences and silence the expression of CRISPR-cas genes [125]. Phages found to contain anti-CRISPRs, during characterisation, may be more applicable for use a therapeutic formulation to prevent CRISPR-cas resistance.
To prevent RM-systems, phages modify their nucleotide bases. Modified nucleotides in phage genomes such as archeosine [54], queosine [126], hypermodified thymidine and guanine bases [127] prevent DNA motif recognition and degradation. During characterisation phages can be identified that carry genes capable of modifying nucleotides and therefore are more resistant to RM-resistance [54]”.
Point 10: line 327/328 ….and supresses the PROPAGATION (instead of evolution) of receptor mutations [105, 106] and of CRISPR-cas resistance, RESPECTIVELY…..
Response 10: We have now amended the sentence accordingly (lines 303-305).
Point 11: line 335 …a solution for those PATIENTS with CF where….
Response 11: We thank to reviewer for this suggestion. However, in consultation with our CF consumer advisory group, they have verbalised issue being described solely as “patients”. They believe it is somewhat stigmatising, so refrain from using this term unless in the context when they have been admitted to hospital. In the place of ‘patients’ the words ‘individuals’ has been used, which a term the advisory group approves of (line 335).
Point 12: Line 337…phage arise, they can also COUNTER-evolve to overcome….
Response 12: We now have amended the sentence accordingly as suggested (line 335).
Point13: line 485-488 the two references # 51 and # 52 are identical, PLEASE CHECK
Response 13: We thank the reviewer for finding this duplication. We have amended the text (lines 129 and 133) and removed the additional duplication.
Reviewer 3 Report
The manuscript presents an extremely interesting issue; Pseudomonas aeruginosa Resistance to Bacteriophages and its Prevention by Strategic Therapeutic Cocktail Formulation. I have read with great attention and interest the whole manuscript. Unfortunately, there are numerous errors and study gaps that make the manuscript unpublishable.
- The manuscript is poorly written.
- All informations are not linked in the same context.
- There is very limited approach in the whole manuscript with few trusted references
I do believe the manuscript should be rejected as such. Considering its potential, I encourage the Authors to update their review paper with more valuable linked informations, they must talk about something new. At current stage they might be used for a conference communication and as a burning spot for further studies. Looking forward for great research articles from your group.
Accept my kind regards
Author Response
Point 1: The manuscript presents an extremely interesting issue; Pseudomonas aeruginosa Resistance to Bacteriophages and its Prevention by Strategic Therapeutic Cocktail Formulation. I have read with great attention and interest the whole manuscript. Unfortunately, there are numerous errors and study gaps that make the manuscript unpublishable.
- The manuscript is poorly written.
- All informations are not linked in the same context.
- There is very limited approach in the whole manuscript with few trusted references
I do believe the manuscript should be rejected as such. Considering its potential, I encourage the Authors to update their review paper with more valuable linked informations, they must talk about something new. At current stage they might be used for a conference communication and as a burning spot for further studies. Looking forward for great research articles from your group.
Response 1: We thank the reviewer for their time in reading our manuscript and providing their critique. There is a resurging interest in this topic and the field of phage therapy (even conveyed by this reviewer) which is why the content is still topical for publication. This essential criterion to justify publication has been acknowledged by all other (3) reviewers. The topic although not extremely novel is being revisited at a time when antibiotic resistance has reach crisis point globally due to the fact that there is no investment into new antibiotics and alternative are still in experimental stage. The fact is phage therapy remains the most translatable alternative therapeutic pipeline for the clinical setting. However, many remain ignorant of the area, its potential use, the challenges that still remain in this translation, as well as concerns in data interpretation. We totally agree that caution is required in interpreting many of the articles in the field, however, this review is obliged to identify the work attempted but more importantly highlight the need for additional rigorous studies that are essential to translate phage therapy into standard clinical care. In response to all reviewers comments, we have extensively improved article, and included several new topics of discussion, appropriate citations and valid interpretation. We feel that combined, the resultant article is publishable and of interest to those in the field. We thank the reviewer for their enthusiasm in our research and the topic.
Reviewer 4 Report
Authors present a review on resistance of clinically relevant Pseudomonas aeruginosa to bacteriophages and discuss an application of the therapeutic cocktails of bacteriophages active against cystic fibrosis-related Pseudomonas aeruginosa variants. The summarized data are important for understanding on applicability of this group of bacteriophages in real practice. In addition, authors highlighted challenges related to a broader application of phage therapy, particularly, against Pseudomonas aeruginosa cells, however several issues should be addressed to accept the manuscript for publishing.
The review should be amended by a critical analysis of the state-of-the-art strategies for development of bacteriophage-based therapies directed towards cystic fibrosis-related Pseudomonas aeruginosa (for example, presented in: Dunne et al. Cell Reports 2019, 29, 1336; Lenneman et al. Current Opinion in Biotechnology, 2021, 68, 151; Gordillo Altamirano, F. L., & Barr, J. J. (2019). Phage therapy in the postantibiotic era. Clinical Microbiology Reviews, 32(2). doi:10.1128/cmr.00066-18).
In addition, the production and a quality control of the bacteriophage preparations as well as challenges to produce therapeutic bacteriophages using the pathogens should be discussed at least in a short paragraph (some suggestions for references: Fernandez et al. The perfect bacteriophage for therapeutic applications—A quick guide. Antibiotics 2019, 8, 126; Jurac et al. Bacteriophage production processes. Applied Microbiology and Biotechnology (2019) 103: 685; Regulski et al. Bacteriophage Manufacturing: From Early Twentieth-Century Processes to Current GMP. In: Bacteriophages, D. Harper et al. (eds.), Springer International Publishing AG 2018, https://doi.org/10.1007/978-3-319-40598-8_25-1).
The reported clinical trials using bacteriophages against Pseudomonas spp. should be also discussed (some suggestions for references: Wright et al. Controlled clinical trial of a therapeutic bacteriophage preparation in chronic otitis due to antibiotic-resistant Pseudomonas aeruginosa; a preliminary report of efficacy. Clin. Otolaryngol. 2009, 34, 349–357; Jault et al. Efficacy and tolerability of a cocktail of bacteriophages to treat burn wounds infected by Pseudomonas aeruginosa (PhagoBurn): A randomised, controlled, double-blind phase 1/2 trial. Lancet Infect. Dis. 2019, 19, 35–45) .
Regarding the bacteriophages harboring the modified nucleotides, not only methylation but also other modifications found in Pseudomonas attacking viruses have to be mentioned (for example, Flores et al. Archives of Virology 2017,162, 2345).
Author Response
Point 1: The review should be amended by a critical analysis of the state-of-the-art strategies for development of bacteriophage-based therapies directed towards cystic fibrosis-related Pseudomonas aeruginosa (for example, presented in: Dunne et al. Cell Reports 2019, 29, 1336; Lenneman et al. Current Opinion in Biotechnology, 2021, 68, 151; Gordillo Altamirano, F. L., & Barr, J. J. (2019). Phage therapy in the postantibiotic era. Clinical Microbiology Reviews, 32(2). doi:10.1128/cmr.00066-18).
Response 1: We thank the reviewer for their recommendation and now have included a paragraph to include the suggested information (lines 94-95 and 104-120).
Point 2: In addition, the production and a quality control of the bacteriophage preparations as well as challenges to produce therapeutic bacteriophages using the pathogens should be discussed at least in a short paragraph (some suggestions for references: Fernandez et al. The perfect bacteriophage for therapeutic applications—A quick guide. Antibiotics 2019, 8, 126; Jurac et al. Bacteriophage production processes. Applied Microbiology and Biotechnology (2019) 103: 685; Regulski et al. Bacteriophage Manufacturing: From Early Twentieth-Century Processes to Current GMP. In: Bacteriophages, D. Harper et al. (eds.), Springer International Publishing AG 2018, https://doi.org/10.1007/978-3-319-40598-8_25-1).
Response 2: In response to this recommendation, the article has been amended to include the suggested information and citations (lines 95-103 and 121-128). Specifically, we felt a dedicated section was warranted and have amended the other sectional numbers accordinly.
Point 3: The reported clinical trials using bacteriophages against Pseudomonas spp. should be also discussed (some suggestions for references: Wright et al. Controlled clinical trial of a therapeutic bacteriophage preparation in chronic otitis due to antibiotic-resistant Pseudomonas aeruginosa; a preliminary report of efficacy. Clin. Otolaryngol. 2009, 34, 349–357; Jault et al. Efficacy and tolerability of a cocktail of bacteriophages to treat burn wounds infected by Pseudomonas aeruginosa (PhagoBurn): A randomised, controlled, double-blind phase 1/2 trial. Lancet Infect. Dis. 2019, 19, 35–45).
Response 3: We thank the author for their suggestions and these aspects have now been added to the manuscript, lines 138-147, “However, in order to be more widely translated clinical trials are required. There have been two clinical trials of phage therapy, for the treatment of P. aeruginosa ear [85] and burn infections [86]. Phage therapy was shown to be effective and safe in the relatively small trial against P. aeruginosa ear infections [85]. Interestingly, in the trial for the treatment of burn wound infections only safety was demonstrated since there was issues manufacturing GMP quality phage cocktails at sufficient titre [86]. Inconsistent results necessitate additional clinical trials not only specifically in this context but for the treatment of other infections such as P. aeruginosa CF lung infections. To enable clinical trials and for phage therapy to become a standard adjunct to antimicrobials in CF, further detailed pre-clinical studies are needed.”
Point 4: Regarding the bacteriophages harboring the modified nucleotides, not only methylation but also other modifications found in Pseudomonas attacking viruses have to be mentioned (for example, Flores et al. Archives of Virology 2017,162, 2345).
Response 4: We thank the author for their recommendation and additional information has been added to the phage evolution paragraph (lines 328-332).
Round 2
Reviewer 3 Report
Dear authors of the manuscript, I can notice the intensive work added to the manuscript. The current version is much better with more clear information that would be useful in this approache.
I accept the current version of the manuscript after more language improvement.
Accept my kind regards
Reviewer 4 Report
The authors addressed all risen points and amended the manuscript with the appropriate references.